# Digital Gene Expression Analysis of Epithelioid and Sarcomatoid Mesothelioma Reveals Differences in Immunogenicity

**DOI:** 10.3390/cancers13081761

**Published:** 2021-04-07

**Authors:** Luka Brcic, Alexander Mathilakathu, Robert F. H. Walter, Michael Wessolly, Elena Mairinger, Hendrik Beckert, Daniel Kreidt, Julia Steinborn, Thomas Hager, Daniel C. Christoph, Jens Kollmeier, Thomas Mairinger, Jeremias Wohlschlaeger, Kurt Werner Schmid, Sabrina Borchert, Fabian D. Mairinger

**Affiliations:** 1Diagnostic and Research Institute of Pathology, Medical University of Graz, 8010 Graz, Austria; luka.brcic@medunigraz.at; 2Institute of Pathology, University Hospital Essen, University of Duisburg Essen, 45147 Essen, Germany; alexander.mathilakathu@stud.uni-due.de (A.M.); Robert.Walter@rlk.uk-essen.de (R.F.H.W.); Michael.Wessolly@uk-essen.de (M.W.); elena.mairinger@ruhrlandklinik.uk-essen.de (E.M.); daniel.kreidt@stud.uni-due.de (D.K.); Julia.Steinborn@uk-essen.de (J.S.); thomas.hager@uk-essen.de (T.H.); KW.Schmid@uk-essen.de (K.W.S.); Sabrina.Borchert@rlk.uk-essen.de (S.B.); 3Department of Pulmonary Medicine, University Hospital Essen—Ruhrlandklinik, 45239 Essen, Germany; hendrik.beckert@rlk.uk-essen.de; 4Department of Medical Oncology, Evang. Kliniken Essen-Mitte, 45136 Essen, Germany; D.Christoph@kliniken-essen-mitte.de; 5Department of Pneumology, Helios Klinikum Emil von Behring, 14165 Berlin, Germany; jens.kollmeier@helios-kliniken.de; 6Department of Tissue Diagnostics, Helios Klinikum Emil von Behring, 14165 Berlin, Germany; thomas.mairinger@helios-gesundheit.de; 7Department of Pathology, Diakonissenkrankenhaus Flensburg, 24939 Flensburg, Germany; wohlschlaegerje@diako.de

**Keywords:** pleural mesothelioma, gene expression, immunogenicity, sarcomatoid, epithelioid

## Abstract

**Simple Summary:**

Malignant pleural mesothelioma (MPM) is a rare, biologically extremely aggressive tumor with an infaust prognosis. In this retrospective study, we aimed to assess the role of tumor-infiltrating immune cells and their activity in the respective histologic subtypes. We confirmed a substantial difference between epithelioid and sarcomatoid mesothelioma regarding the host’s anti-cancer immune reaction. Whereas antigen processing and presentation to resident cytotoxic T cells as well as phagocytosis is highly affected in sarcomatoid mesothelioma, cell–cell interaction via cytokines seems to be of greater importance in epithelioid cases. Our work reveals the specific role of the immune system within the different histologic subtypes of MPM, providing a more detailed background of their immunogenic potential. This is of great interest regarding therapeutic strategies addressing immunotherapy in mesothelioma.

**Abstract:**

Malignant pleural mesothelioma (MPM) is an aggressive malignancy associated with asbestos exposure. Median survival ranges from 14 to 20 months after initial diagnosis. As of November 2020, the FDA approved a combination of immune checkpoint inhibitors after promising intermediate results. Nonetheless, responses remain unsatisfying. Adequate patient stratification to improve response rates is still lacking. This retrospective study analyzed formalin fixed paraffin embedded specimens from a cohort of 22 MPM. Twelve of those samples showed sarcomatoid, ten epithelioid differentiation. Complete follow-up, including radiological assessment of response by modRECIST and time to death, was available with reported deaths of all patients. RNA of all samples was isolated and subjected to digital gene expression pattern analysis. Our study revealed a notable difference between epithelioid and sarcomatoid mesothelioma, showing differential gene expression for 304/698 expressed genes. Whereas antigen processing and presentation to resident cytotoxic T cells as well as phagocytosis is highly affected in sarcomatoid mesothelioma, cell–cell interaction via cytokines seems to be of greater importance in epithelioid cases. Our work reveals the specific role of the immune system within the different histologic subtypes of MPM, providing a more detailed background of their immunogenic potential. This is of great interest regarding therapeutic strategies including immunotherapy in mesothelioma.

## 1. Introduction

Malignant pleural mesothelioma (MPM) is a rare type of cancer that is heavily associated with asbestos exposure [1,2]. This malignancy originates from the pleural mesothelium and is associated with a bad prognosis. Median survival times range from 14–20 months after initial diagnosis [3,4,5]. Generally, MPM can be differentiated into three major histologic subtypes, epithelioid (EMM), sarcomatoid (SMM), and biphasic (BMM). EMM accounts for up to 80% of all MPM cases [6]. It has also a more favorable outcome compared with the SMM or BMM, especially when surgery is applied [7]. Though it needs to be noted that epithelioid morphology can differ greatly [6,8], thereby also impacting clinical outcome [9,10,11]. The sarcomatoid subtype is the least prevalent subtype of mesothelioma (<10%) [8]. SMM is considered to be more aggressive in a clinical setting with a higher tendency of distant metastasis [6,12]. The BMM has a mixed composition of both epithelioid and sarcomatoid histology [8]. It is currently discussed whether a proportion of specific histology in biphasic MPM has a prognostic value [13,14].

As distinct biomarkers are lacking [15], early detection is often impeded, thereby worsening patients’ outcome. Unfortunately, only a small fraction of patients is suitable for pleurectomy [16], while most patients are treated with a cisplatin/pemetrexed combination. The treatment may prolong overall survival by 3 months [5]. Meanwhile, patients undergoing palliative care including palliative chemotherapy may have an overall survival of 9 months. Immune checkpoint inhibitors are also used as a treatment option in MPM. These inhibitors target negative regulatory immune checkpoints on immune cells, thereby enhancing a prevalent immune response against the tumor. Single agents (pembrolizumab, a PD-1 inhibitor) have shown increased response rates; however, they have failed to show benefits for progression-free (PFS) or overall survival (OS) [17]. Despite this setback, the Checkmate 743 study revealed a four-month OS benefit (mOS, HR: 0.74, CI: 0.60–0.91, *p*-value: 0.0020) and increased two-year survival rate (41% vs. 27%), when comparing immune checkpoint doublet therapy (ipilimumab and nivolumab) with standard systematic chemotherapy [4]. Nonetheless, responses remain unsatisfying with only marginal improvements compared to the best supportive care [18]. With immune therapy now in the focus of current mesothelioma treatment, a deeper knowledge of the tumor’s immunogenic potential may help to improve patient selection for this form of therapy.

Though the immune system is widely recognized for its anti-tumor activity, it plays a dual role in MPM and may also support tumor survival and progression. Inhaled microfibres, which are released during processing, corrosion, and weathering of asbestos, often reside in pleural tissue. Unfortunately, macrophages are unable to decompose them [3]. Over time, the persistent fibers damage adjacent cells, leading to necrosis and potentially triggering an immune response. The resulting chronic inflammatory reaction can induce tumor mutagenesis via release of reactive oxygen species (ROS) [19]. These macrophages, together with other various not-tumor-derived cell types essential for MPM development, constitute the so-called tumor microenvironment (TME) [20]. Three important immune cell types, known to infiltrate MPM, are tumor-associated macrophages (TAMs), T-lymphocytes, and myeloid-derived suppressor cells (MDSCs) [20]. TAMs are generally considered to be the largest subset of cells infiltrating MPM (up to 42%) [21,22]. Non-tissue resident macrophages are attracted to the tumor site via expression of the chemokine CCL2 [23]. Once within the tumor, growth factors expressed by the tumor (M-CSF, IL-34, TGF-b, and IL-10) induce an immunosuppressive macrophage phenotype (M2 macrophages) [23,24,25]. From a clinical perspective, the immune suppressive effects of macrophages are associated with poor prognosis and resistance to standard chemotherapy [23]. Some studies suggested macrophage-based biomarkers to estimate prognosis and outcome in EMM [26,27,28]. Despite next-generation sequencing studies identifying few mutations resulting in presented neoepitopes and increased immunogenicity [29], T-lymphocytes are the second biggest fraction of the immune cell infiltrate (20–42%), closely following TAMs [27,30,31]. It is speculated that the neoepitope load is higher than suggested, as chromosomal rearrangements can not be detected by targeted amplicon-based NGS, which are often present in MPM [32]. The infiltrating lymphocytes are mostly CD8-positive cytotoxic T lymphocytes (CTL), as well as CD4 and FoxP3 positive regulatory T cells (Tregs) [22,31]. Strikingly, based on pleural effusions of MPM, regulatory T-cells are less common when compared to other tumor entities [25]. Though high infiltration rates and activity of CTL are observed in MPM [25,33], they display signs of anergy or exhaustion [34]. MDSCs are the smallest fraction of the immune cell infiltrate (up to 9%) [30,35]. These cells are predominately associated with suppression of T-cells via releasing of ROS and PD-L1 expression [35,36,37]. Furthermore, a higher concentration of MDSCs can be linked to poor prognosis in EMM [27,38]. Based on these findings one can conclude that the majority of acting immune cells at the tumor site are either ineffective or are reprogrammed to support tumor growth and progression. Unfortunately, most studies did not distinguish between EMM and SMM when analyzing tumor immune infiltration or are only based on limited numbers of SMM samples. A recent study showed the infiltration of CD8+ T cells as being twice as high in SMM than in EMM but included only six SMM [39].

The above-mentioned points highlight the importance of the immune system for MPM development and progression and raise the question of how different immunogenicity contributes to the different outcomes between EMM and SMM. Deepening the understanding of the biological background of immune escape mechanisms in those histologic subtypes might carry the potential for new therapeutic approaches and improved clinical management of patients in the future.

## 2. Materials and Methods

### 2.1. Patient Cohort and Experimental Design

This retrospective study was performed on therapy-naïve, formalin-fixed paraffin-embedded samples of 22 patients with MPM treated at the West German Cancer Centre or the West German Lung Centre (Essen, Germany) between 2006 and 2009 and the Helios Klinikum Emil von Behring (Berlin, Germany) between 2002 and 2009. Twelve of those were diagnosed as SMM and 10 as EMM. The diagnosis was confirmed by two experienced pathologists (JWO, KWS), based on the current WHO classification [40]. Patients were staged according to the 2017 UICC/AJCC staging [41]. Inclusion criteria were the availability of sufficient tumor material and a complete set of clinical data concerning follow-up and treatment. All patients received platinum-based chemotherapy. The radiologic response rate was assessed by modified Response Evaluation Criteria in Solid Tumours (modRECIST) [42]. Surveillance for this study was stopped on August 31, 2014. Complete follow-up was available for all patients with reported deaths of all patients. Clinicopathological data of the study cohort are summarized in Table 1.

### 2.2. RNA Isolation and Integrity Assessment

RNA was purified from 20 µm thick FFPE sections, using the Maxwell RSC RNA FFPE Kit supplied by Promega. Obtained RNA was eluted in 50 µL RNase-free water and stored at −80 °C. Before the assessment, RNA concentration was determined via Qubit Fluorometric Quantification (Thermo Fisher Science, Waltham, MA, USA) undergoing manufacturer’s instructions for the RNA broad range assay kit. Ultimately, 200 ng of each sample was processed.

### 2.3. Digital Gene Expression Analysis

For evaluation of the RNA expression pattern, the commercially available NanoString PanCancer Immune Profiling Panel including 770 immune-related as well as 30 reference genes was used. All code sets along with experiment reagents were designed and synthesized by NanoString Technologies (Seattle, WA, USA). The post-hybridization processing was performed using the nCounter MAX/FLEX System (NanoString) and cartridges were scanned on the Digital Analyzer (NanoString). Samples were analyzed on the NanoString nCounter PrepStation, using the high-sensitivity program, and cartridges were read at maximum sensitivity (555 FOV).

### 2.4. NanoString Data Processing

NanoString data processing was performed with the R statistical programming environment (v4.0.2) using NanoStringNorm [36] and NAPPA package, respectively. Considering the counts obtained for positive control probe sets, raw NanoString counts for each gene were subjected to a technical factorial normalization, carried out by subtracting the mean counts plus two-times standard deviation from the CodeSet inherent negative controls. Afterward, a biological normalization using the geometric mean of all reference genes was carried out. To overcome basal noise, all counts with *p* > 0.05 after one-sided *t*-test versus negative controls plus 2× standard deviations were interpreted as not expressed.

### 2.5. Statistical Analysis

Statistical analysis was carried out using the R statistical programming environment V 4.0.2. Prior to exploratory data analysis, the Shapiro–Wilks-test was applied to test for normal distribution of each dataset for ordinal and metric variables. The resulting dichotomous variables underwent either the Wilcoxon Mann–Whitney rank sum test (non-parametric) or the two-sided student’s *t*-test (parametric). For comparison of ordinal variables and factors with more than two groups, either the Kruskal–Wallis test (non-parametric) or ANOVA (parametric) were used to detect group differences.

Double dichotomous contingency tables were analyzed using Fisher’s exact test. To test the dependency of ranked parameters with more than two groups the Pearson’s Chi-squared test was used. Correlations between metrics were tested applying Spearman’s rank correlation test as well as Pearson’s product-moment correlation testing for linearity.

Basic quality control of run data was performed by mean-vs-variances plotting to find outliers in target or sample level. True differences were calculated by correlation matrices analysis. Pathway analysis is based on the KEGG database and was performed using the “pathview” package of R. Differences were specified by −log2 fold changes between means (if parametric) or medians (if non-parametric) of compared groups. Significant pathway associations were identified by gene set enrichment analysis using the WEB-based GEne SeT AnaLysis Toolkit (WebGestalt) [43,44,45]. Each run was executed with 1000 permutations. Finally, all associations were ranked according to the false discovery rate (*p* < 0.05).

Due to the multiple statistical tests, the *p*-values were adjusted by using the false discovery rate (FDR). The level of statistical significance was defined as *p* ≤ 0.05 after adjustment.

## 3. Results

### 3.1. Gene Expression Pattern of Immune-Related Genes

Overall, 304 out of 698 (43.6%) significantly expressed immune-related genes show differential expression between EMM and SMM, indicating an overall difference in interaction with the host’s immune system. In particular, 90 of those 304 genes (29.6%) show expression only or in a much stronger manner in SMM compared to EMM cases, whereas 214 targets (70.4%) present with overexpression in EMM. In ranked order, ABCB1, SYCP1 und IFNA7 show most differences between both subtypes, with solid expression levels (between about 500 counts for SYCP1^ and up to nearly 3000 counts for IFNA7) in EMM but an absence of expression in SMM, whereas MAPK8, AXL und UBC show gene expression predominantly in sarcomatoid cases.

No differences in infiltration density of CD8+ CTL could be observed (FDR adj. *p* = 0.901). Of note, CD4+ T-cells, as well as CD68+ macrophages, were enriched in the SMM. CD20+ B cells tend to be denser in EMM than in SMM, but the overall expression of MS4A1 (CD20) is only slightly above background (20 vs. 100 counts in median) and the association did not reach statistical significance after adjustment (*p*-value: 0.050; FDR adj. *p*-value: 0.094).

An overview of all differences in gene expression pattern between the two histologic subtypes is illustrated in Figure 1, an overview of all *p*-values and statistical parameters can be found in Appendix A.

### 3.2. Gene Set Enrichment Analysis (GSEA)

To identify biological background mechanisms (pathways and biological functions/categories) behind the different expression patterns regarding immune-related genes in EMM and SMM, a Gene Set Enrichment Analysis (GSEA) was performed (Figure 2).

In the SMM mainly the pathways for phagosome, antigen processing and presentation, lysosome, autoimmune thyroid disease, viral myocarditis, Fc gamma R-mediated phagocytosis, Eppstein–Barr virus infection, endocytosis, focal adhesion, and proteoglycans in cancer show the strongest enrichment. On the other hand, cytokine–cytokine receptor interaction, salmonella infection, inflammatory mediator regulation of TRP channels, adrenergic signaling in cardiomyocytes, amoebiasis, African trypanosomiasis, parathyroid hormone synthesis, secretion and action, NF-kappa B signaling pathway, inflammatory bowel disease, and Kaposi sarcoma-associated herpesvirus infection are identified as enriched and thereby potentially activated in EMM.

Details of the GSEA, including normalized enrichment score, the *p*-value of enrichment, exact targets included in the gene sets, and those differentially regulated, can be found in Appendix A.

The main altered/influenced pathways are described in particular in the following paragraphs:

#### 3.2.1. Phagocytosis and Antigen Presentation

All phagocytosis- and antigen-presentation associated signaling pathways, including phagosome (Appendix A), antigen processing and presentation (Appendix A), lysosome, Fc gamma R-mediated phagocytosis (Appendix A), and endocytosis are strongly enriched in SMM. For direct phagocytosis, this includes important factors involved in the phagolysosome, like LAMP or cathepsin β, antigen processing and cross-presentation, like TAP1/2 or MHC I/II molecules, or the cytochrome b558 mediated activation of NADPHoxidase, with strong overexpression of gp91 and p40phox. Furthermore, strong expression levels of most phagocytosis-promoting receptors, including Fc receptors, complement receptors, integrins, toll-like receptors, C-lectin receptors as well as Scavenger receptors, could be shown. Accumulation of CD45 positive cells, as activators of T cell response, expression of the Fcγ receptors FcγIIA and B, and downstream signaling via Src and Syk could be verified. Besides antigen processing via autophagy, the “classic” proteasome-associated mechanism for antigen processing and presentation via TAP1/2, TAPBP, and MHC1 binding showed strong activation on all levels of the MHC I pathway for antigen presentation to CD8+ CTL and KIR+ NK cells. Furthermore, the MHC II pathway, important for antigen presentation to CD4+ helper T-cells via MHC II, is overexpressed in total, including but not limited to Ii, MHC2, SLIP, CTSB/L/S, CLIP, and HLA-DM.

#### 3.2.2. Cell–Cell Interaction and Communication within the Tumor Microenvironment

MPM subtypes show a clear difference in the communication networks used between the tumor cells and/or different immune cell types. This spans biological mechanisms and pathways from cytokine–cytokine receptor interactions over cell–cell interaction via proteoglycans up to differences in focal adhesion (Appendix A). This could be shown by highly increased expression levels of hyaluronan (HA, including CD44, CD44v3), heparan sulfate proteoglycans (HSPGs, including the integrins α2β1, avβ3 or α5β1 and fibronectin) as well as chondroitin/dermatan sulfate proteoglycans (CSPG/DSPG, including TLR2 and TLR4) (Appendix A).

For cell communication via cytokines, especially γ-chain utilizing class I helical cytokine receptors (IL2RA, IL2RG, IL4R, IL15RA, IL21R, IL7R) and IL4-like receptors (IL3RA, CSF2RB, IL13RA1), significantly elevated gene expression in SMM compared to EMM was shown. In EMM samples, an enrichment of IL6/12-like (IL6R, IL11RA, IL12RB2) and IL1-like receptors (IL1R2, IL1RL2, Il18R1, ST2) could be observed.

On the side of chemokine secretion, markable differences in CXC subfamily member expression was observed, whereas those binding CXCR1 (CXCL1, CXCL5, CXCL6) and CXCR2 (CXCL2, CXCL3, CXCL7) are expressed in EMM and those binding CXCR3 (CXCL9, CXCL10, CXCL11) or CXCR5 (CXCL13) are expressed in SMM (Appendix A).

## 4. Discussion

For a long time, tumors have been widely underestimated in their complexity, viewed as a clustering of cancer cells on their own, and not considered in terms of the importance of extracellular signaling and complex interactions in the TME. Since then, extensive research has been conducted on the topic of tumor-associated immune events, revealing their enormous influence on tumor progression. In this study, we have approached MPM as a cancer entity with an especially heterogenous TME, whose composition might also be of prognostic value [46]. Our data analysis revealed numerous factors and pathways involved in the cell cycle progression, presumably acting in a synergistic effect and offering an explanation for the progression of MPM despite therapy.

### 4.1. Phagocytosis

Despite the understanding of the decisive role the phagosome pathway plays in cancer, it has not yet been described for MPM. GSEA in our study revealed the following phagocytotic pathways being affected with high significance: phagosome, Fc gamma R-mediated phagocytosis, lysosome, and endocytosis. As the phagosome pathway showed the highest enrichment (2.5), we focused on differences between gene expression of selected SMM and EMM genes in this pathway (Appendix A). The phagosome pathway is mainly involved in the response of the innate immune defense and includes endocytosis, phagocytosis, phagosome maturation, and the development of the lysosome [47]. Phagocytes (macrophages, granulocytes, or dendritic cells) use their plasma membrane to engulf a large particle (e.g., apoptotic cell or microbes) [47]. Tumor cells are also engulfed by phagocytes. The ensuing early endosome fuses with the lysosome into a late endosome, then diffused through the membrane of the phagolysosome. Cathepsins are key acid hydrolases within the lysosome. They are associated with the processes of the lysosome, including the process of antigen presentation [48]. Cathepsins represent the principal effectors of protein catabolism and autophagy and support the increased metabolic needs of proliferating cancer cells [48]. In this study, cathepsin was overexpressed in SMM. Overexpression of cathepsin is associated with poor prognosis [48,49]. LAMPs were also overexpressed in SMM. This family of glycosylated proteins is involved in supporting tumor growth and metastatic spread [50].

Toll-like receptors (TLRs) are involved in the response of the innate immunity, but can also organize several downstream signaling pathways leading to the formation or suppression of cancer cells [51]. Once synthesized, they are translocated to the Golgi complex and subsequently delivered to the plasma or endosomes [51]. Overexpression of TLRs has been reported for several cancers like prostate cancer, neuroblastoma, lung cancer, and ovarian cancer. While in some studies overexpression of TLRs has been associated with more aggressive forms of, e.g., squamous cell carcinoma [52], other studies revealed high expression being indicative of longer survival rates [53]. In our study, in contrast to SMM with increased expression of TLR2 and TLR4, EMM exhibited overexpression of TLR6. TLR6 is suggested to have an anticancer function, as described in the literature for colon cancer [54]. TLR2 and TLR4 have been associated with gastric cancer [55].

The TAP transporter and MHC class I and II molecules are involved in the process of antigen processing and cross-presentation. These are overexpressed in phagocytes of SMM. As these molecules are also involved in antigen processing and presentation, this finding is further discussed in Section 3.2.

### 4.2. Antigen Processing and Presentation

Modern immunotherapeutic approaches have already been investigated in clinical trials in MPM [56,57,58]. One possible explanation for different responses might be in the processing and presentation of tumor-specific epitopes [59,60] important for the activation of tumor-specific T-cells [61]. A complex intracellular pathway is involved in processing these antigenic peptides (Appendix A). It starts with the polyubiquitination of the protein, which is then degraded by the proteasome. We have previously demonstrated strong 20S proteasome expression in MPM [62]. Its function is to remove misfolded/dysfunctional proteins, but high expression might lead to an “overheated” proteasome with deficient antigen processing capabilities. This could explain why the high expression of proteasomal components is associated with worse outcomes in MPM [62]. Translocation of small fragments processed by the proteasome into the endoplasmatic reticulum is performed via the TAP-transporter, a homodimer composed of TAP1 and TAP2 [63]. These peptide fragments bind the HLA class I molecule, and the whole complex is transported to the cell surface where it is recognized by CTL [61]. Classically, three genes (HLA-A, HLA-B, HLA-C) with an ample number of alleles code for the HLA class I molecule, but inferior genes are also known [64]. In the present study, we demonstrated a markable upregulation of gene expression levels of the above-mentioned components in SMM. Elevated CD68 expression levels (higher amount of macrophages) increased the activation of antigen-presentation-associated pathways in macrophages and dendritic cells with simultaneously even levels of CD8+ CTL, and no signs of direct anti-cancer immune aggression (like an expression of perforin or granzymes), implies altered processing of tumor neoantigens. This results in a “last-ditch attempt” of antigen-presenting cells to stimulate cytotoxic lymphocytes and NK cells. Deficiencies of the antigen presentation resulting in immune evasion from CTL are well described in different tumors [65,66]. These include the deficiency of HLA/MHC class I molecules due to point mutations or large deletions, but also mutations in HLA/MHC class I subunits, like β-2 microglobulin [56]. Furthermore, tumors might be capable of regulating HLA/MHC class I expression on an epigenetic level via DNA hypermethylation [67]. Johnsen et al. observed the development of large and persistent tumors through TAP1-negative parental transformed murine fibroblast cell line. In the case of tumor progression, TAP1-negative cells have been reported to be selection-wise favored over TAP1-positive cells [68]. Already in 1993, Restifo et al. suggested a possible tumor escape mechanism through deficient antigen presentation and processing based on finding of low mRNA levels for LMP-2 and LMP-7 (proteasome subunits) and TAP1 and TAP2 in small lung cell carcinomas [69]. Additional escape mechanisms involving TAP-mutations and cofactors that interact with TAP have been described [63]. The missing potency of cytotoxic T lymphocytes activity against the tumor cells by altered antigen processing and presentation could explain the inhomogenous response rates in the Checkmate 743.

### 4.3. Proteoglycans in Cancer

In recent decades, extracellular matrix (ECM) and TME have been recognized as major factors of tumor development and progression. In ECM, many different proteins and molecules are regulating different processes important for carcinogenesis. One of the key players in ECM is fibronectin (FN), which was found to be overexpressed in SMM in this study. FN is a glycoprotein with a central role in tumor cell proliferation, angiogenesis, invasion, and metastasis development, but also in processes involved in tumor evasion of the immune system (for review see [70]). Furthermore, its overexpression in SMM is not surprising, since FN is an important mesenchymal marker, and when found in epithelial malignancies is used as a sign of epithelia-mesenchymal transition (EMT) [71]. Its activation of TGF-β induces a partial EMT phenotype, usually at the invasive front of the epithelial tumors [72]. We have also found increased expression of integrin receptors α5 β1, α2 β1 and αv β3 in SMM in our cohort. Integrins are cell adhesion receptors, and the main receptor for ECM proteins and FN, and therefore also involved in many pro-tumor activities like tumor cell proliferation, metastasis, tumor angiogenesis. Binding between FN and integrins is further enhanced by integrin clustering and interacting with urokinase plasminogen activator receptor (uPAR), also overexpressed in SMM [73,74].

Another overexpressed protein in SMM was CD44. CD44 is a transmembrane glycoprotein and primary receptor through which hyaluronan (HA) activates different intracellular pathways resulting in tumor cell growth, migration, invasion, and angiogenesis [75,76]. HA, the only proteoglycan which is not covalently attached to protein core is related to poor prognosis in breast, colon, and ovarian carcinoma [77], and its presence in tumor stroma is an indication of the more aggressive tumor [78,79,80]. It has been shown that HA in MPM is overexpressed in intracellular, but also in pleural, fluid [81]. Hanagiri et al. demonstrated that the interaction of HA with CD44 is important for the proliferation and migration of tumor cells in MPM [82]. Interestingly, overexpression of CD44 was not observed in the EMM group.

As previously mentioned, we have also found overexpression of TLR 2 and TLR4, which are receptors for decorin, proteoglycan important for growth control, usually with binding and inactivation of TGF-β [83,84,85], inhibition of angiogenesis, and inducing of apoptosis through EGFR down-regulation [86]. It has been shown that decorin, through TLR2 and TLR4, induces proinflammatory tumor suppressor programmed cell death 4 (PDCD4), whose degradation is further prevented through the TGF-β1 blockade [87].

Thrombospondin-1, overexpressed in SMM, is a very controversial ECM protein involved in cell survival, migration, invasion, angiogenesis, and inflammation. However, its role is not straightforward and depends on tumor and ECM type. It is regarded as an anti-angiogenic factor, but some studies have reported its angiogenic activity as well [88]. It was described as a pro-adhesive protein but can also decrease the adhesion of tumor cells and promote invasion and metastases [89,90].

Very similar is the role of lumican, keratan sulfate, in cancer. Its expression is correlated with poor outcome in lung carcinoma, and in colorectal carcinoma, but is a favorable prognostic factor for osteosarcoma and melanoma [91,92,93]. It is known that lumican induces FAS by binding FAS ligands and in this way plays a role in the initiation of apoptosis and suppresses cell proliferation [94,95,96]. FAS is highly expressed in our EMM cohort. At the same time, TGF-β2, which is involved in growth suppression and cell adhesion in osteosarcoma [97], and is negatively regulated by lumican, has been highly expressed in SMM.

### 4.4. Secretion of Cytokines and Communication with the Immune System

To establish themselves and progress properly, it is inevitable for cancer cells to shape their local microenvironment to their benefit. This goal is achieved through continuous inflammatory reactions and heavy modulations of the immune response [98]. With cytokines and out of those especially chemokines being essential mediators for such a process, changes in their expression patterns are of great interest if we are to develop a deeper understanding of MPMs acquired TME. Various ligands, as well as receptors within the CC chemokine subfamily, were overexpressed in both MPM subtypes. This upregulation might support the flourishment of MPM since these chemokines have already been considered to play a vital role in tumor genesis, while their overexpression also appears to modulate the hosts’ immune response against cancer cells [99]. We found a more distinguishable expression pattern regarding the CXC chemokine family. The EMM cases overexpress ligands for CXCR1/2, whereas the sarcomatoid subtype appears to stimulate the CXCR3-pathway with CXCL 9–13. Especially, the activation patterns measured in the EMM stick out, as the CXCR1/2 pathways are thought to contribute massively to the development of, among others, prostate, lung, colorectal, and breast cancer, as well as inflammatory diseases such as COPD and asthma [100,101]. Furthermore, malignancies appear to increase their therapy resistance by overexpression of these receptors and their ligands. In fact, the CXCR1/2 axis has been unrevealed as a potential therapeutic target in malignant melanoma, with pathway-inhibition significantly improving sensitivity for chemotherapy in otherwise resistant melanoma cells in vitro [102], while also decreasing progression and metastasis even in advanced disease [103].

Interleukins are considered to play a key role in MPM development. It was shown that asbestos-exposed knockout mice bearing modified inflammasomes, resulting in a diminished IL-1β release, had a significantly reduced incidence of MPM and later disease onset compared to their wild-type counterparts [104]. Furthermore, IL-6 is thought to not only essentially contribute to MPMs asbestos-related development, but also to impede effective chemotherapy and inducing angiogenesis by increasing VEGF expression [105,106]. In our study, the SMM demonstrated a surprisingly broad spectrum of elevated receptor expressions throughout interleukin 2-, as well as interleukin 4-like receptors. Interestingly, both subtypes, epithelioid via receptor-, sarcomatoid via ligand-upregulation, heavily stimulate the IL-6R pathway.

Especially the recruitment of TAMs has already been considered as a promising therapeutic target in MPM [107]. This hypothesis is further substantiated by Blondy et al., who discovered that MPM cells are directly involved in the recruitment of immunosuppressive macrophages by stimulation of the M-CSF/IL-34/CSF-1R pathway [108]. This perfectly fits the above-mentioned narrative since we were also able to demonstrate an elevated expression of mentioned pathways in our GSEA. Moreover, particularly the SMM upregulates the production of TNF- related TWEAK and TRAIL, as well as TGF-β related ligands TGFB-1 and -2. While the role of TNF has already been established in various malignant processes [109], TGF-ß has even been unraveled as an essential factor in MPM genesis [110,111].

An interesting thought occurred while regarding our expression patterns in the light of modern therapeutic approaches. In a recent study, Horn et al. demonstrated improved immune response and prognostically favorable TME remodeling of breast and lung cancer in a murine model after simultaneous inhibition of the CXCR1/2 and TGF-ß pathway during PDL-1 therapy [112]. As PDL-1 treatment in combination with a cisplatin-pemetrexed based chemotherapy [4,113] has yielded relatively promising results in MPM therapy so far, and with us showing increased activation of the corresponding pathways, transferring this experimental approach to the MPM might be important for future multimodal treatment.

Our study has several technical and biological limitations. As the present study is only based on gene expression data, the final proof of differences in the composition and quantity of the infiltrating immune cells, and chemokine secretion described above is lacking. Furthermore, the relatively small sample sizes of SMM and EMM reduce study strength, as variability, especially between samples of different ethnical origins, may be underestimated. Furthermore, it would be of great interest to analyze the expression of genes involved in innate and acquired immunity in normal mesothelium and compare these findings with EMM and SMM. It is known that normal mesothelial cells form a protective barrier, and are involved in antigen presentation, inflammation, and cell adhesion [114,115]. However, normal pleural tissue from healthy patients can only rarely be provided, which makes it more difficult to characterize a “normal” state and define EMM and SMM-specific features.

## 5. Conclusions

Immune evasion as a hallmark of cancer and both in EMM and SMM can be a problematic issue for therapeutic intervention. Our work reveals the specific gene expression pattern of genes involved in immunological and inflammatory processes within the different histologic subtypes of MPM, providing a more detailed background of their immunogenic potential and demonstrating their distinct pattern of immunogenicity. Those differences comprise genes associated with antigen processing and presentation to resident cytotoxic T cells as well as phagocytosis, but also cell–cell communication via the cytokine system. Knowledge about underlying biological processes has the potential to pave the ground for patient stratification for modern therapeutic approaches such as immune-checkpoint blockades and will be the key for improved clinical management of patients with MPM.

## Figures and Tables

**Figure 1 cancers-13-01761-f001:**
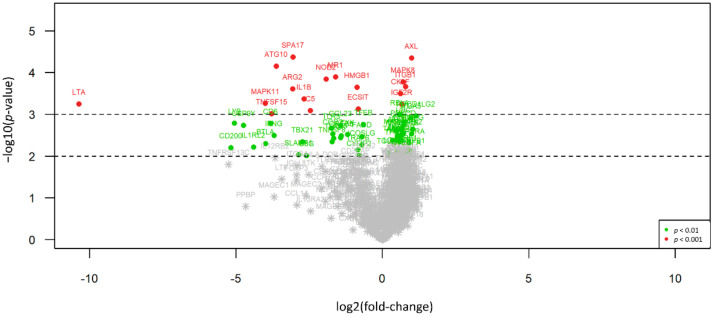
Volcano plot illustrating the differential expression between EMM and SMM. 90 of 304 differentially expressed genes (29.6%) show expression only or in a much stronger manner in SMM (right side) compared to EMM cases, whereas 214 targets (70.4%) present with overexpression in EMM (left side). Red dots indicate highly significant and green dots significant association identified by explorative data analysis using either Wilcoxon Mann–Whitney rank sum test (non-parametric) or the two-sided student’s *t*-test (parametric).

**Figure 2 cancers-13-01761-f002:**
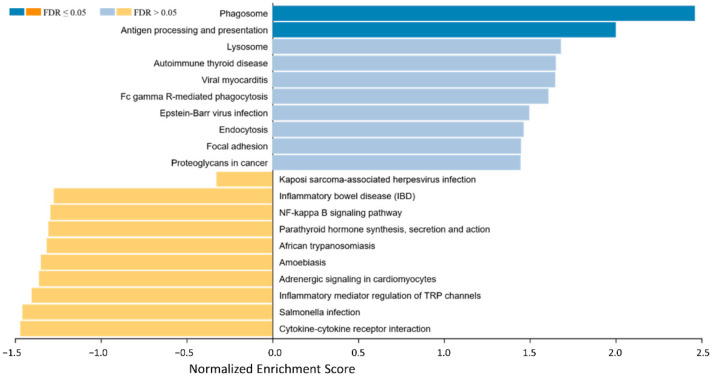
Gene set enrichment analysis of differential expressed genes between EMM and SMMpresenting an overview of gene sets enriched in SMM (right side, blue bars) and EMM (left side, yellow bars). In the SMM the pathways for phagosome, Fc gamma R-mediated phagocytosis, antigen processing and presentation and proteoglycans in cancer show enrichment. Cytokine–cytokine receptor interaction is enriched and thereby potentially activated in EMM.

**Table 1 cancers-13-01761-t001:** Clinicopathological data of the study cohort.

Histology	Age	Sex	T Stage	N Status	M Status	UICC/AJCC	Overall Survival in Months	Outcome	Progression-Free Survival in Months	Initial Progression
EMM	52	M	2	2	0	3B	9.3	Death	5.5	Yes
EMM	56	M	3	0	1	4	43.2	Death	5.5	No
EMM	61	M	2	2	1	4	2.1	Death	1.2	Yes
EMM	65	M	2	2	0	3B	8.8	Death	4.9	Yes
EMM	68	F	2	0	1	4	3.7	Death	3.5	No
EMM	70	M	1	2	0	3B	14.5	Death	6.7	Yes
EMM	73	M	2	0	0	1B	18.0	Death	4.8	Yes
EMM	75	M	3	0	0	1B	21.7	Death	6.4	Yes
EMM	76	M	2	0	0	1B	44.2	Death	14.3	Yes
EMM	77	M	2	0	0	1B	4.6	Death	3.8	Yes
SMM	54	M	2	1	0	2	3.2	Death	2.6	No
SMM	59	M	2	0	0	1B	7.2	Death	7.1	No
SMM	61	m	3	0	0	1B	25.2	Death	11.6	Yes
SMM	62	F	3	1	0	3A	8.9	Death	2.8	Yes
SMM	64	M	3	0	1	4	12.2	Death	5.5	No
SMM	66	M	1	0	0	1A	11.3	Death	9.7	No
SMM	66	M	2	0	1	4	8.4	Death	3.5	Yes
SMM	69	M	4	2	0	3B	8.0	Death	1.4	Yes
SMM	70	M	3	0	0	1B	21.6	Death	11.6	Yes
SMM	71	M	2	2	0	3B	0.8	Death	0.2	No
SMM	79	F	3	2	1	4	13.6	Death	4.1	Yes
SMM	82	M	2	0	0	1B	13.3	Death	9.3	Yes

Legend: EMM—epithelioid malignant mesothelioma, SMM—sarcomatoid malignant mesothelioma.

## Data Availability

All data are available from the author directly.

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
