# Peer review of "Digital Gene Expression Analysis of Epithelioid and Sarcomatoid Mesothelioma Reveals Differences in Immunogenicity"

_cancers, 2021, doi:10.3390/cancers13081761_

Round 1

Reviewer 1 Report

 General comments:

This paper evaluates the difference in gene expression between two histologic subtypes of malignant mesothelioma, that is sarcomatoid and epithelioid. They find significant differences as far as genes involved in either innate or acquired immune defense mechanisms are regarded. The findings they report are undoubtedly of value and deserves to be published. However, apart from the various comments reported below, two main observations should be considered.

1. The normal mesothelium is known to play a key role in defense mechanism, through ingestion of different particles and microorganisms, antigen presentation, ROS production and and cytokine secretion (The International Journal of Biochemistry & Cell Biology 36 (2004) 9–16; Respiration 2008;75:121–133 DOI: 10.1159/000113629. Therefore it is not a surprise that neoplastic cells in SMM could maintain some features of normal cells while EMM do not. Perhaps their absence in EMM may be important. It should be of great interest to analyze the expression of genes involved in innate and acquired defense in normal cells for comparing them with those of SMM and EMM. This aspect could be at least discussed.

2. Conclusions must be limited to underline the differences in gene expression. Suggesting that in SMM, TAM and APC recruitment, phagocytosis and antigen processing and presentation could modulate local immune response is excessively speculative (considering the poor prognosis of this histotype).

The same comment is true for EMM. Therefore a more adequate concluding remarks are required.

Specific comments:

Page 1, line 33

I think that the acronim used for Formalin Fixed Paraffin Embedded (FFPE) should be explained.

If you indicate If you report: twelve of those samples……..please continue to indicate ..ten epithelioid

Page 2, line 57

…setting

Page 2, lines 86-111

In these paragraphs the authors briefly describe the cellular components known to form the MPM micro-environment. However at this point a comparison between the cell types present in SMM and those present in EEM should be important and may also result useful to interpretate the findings. The description as reported suggest that the same cell infiltrate seems to be present in both the MPM histotypes. Only a high level of MDSC   linked to a poor prognosis in EMM is cited.

Page 6, Fig. 2

Fig. 2 is very difficult to analyze. I suggest, if possible to include B-F as single pictures in supplemental materials and leave only A in fig. 2

Page 7, line 243

It is not correct to include TfR among the genes involved in the phagocytic process. The complex Tf-TfR is endocytosed through a chlatrin mediated mechanisms and, following acidification inside the cell, the vesicle recycles to the membrane making free again apotransferrin. Therefore I suggest to omit TfR from the list of genes involved in the ingestion process.

Page 7, line 245

…… NADPH…..

Page 10, lines 406-449

 Resolving the complexity of chemokines network in a simple scheme is very difficult. For example following the chemokine/chemokine receptor guide of 2018 (The FEBS Journal 285 (2018) 2944–2971 ª 2018 ), it appears that CXCR1/2 is be involved in neutrophil recruitment. Are neutrophils recruited in EMM? Neutrophils are very citotoxic cells and this feature could be in principle beneficial for the host.

Conversely CXCR3, accordingly to recent published papers, (Immunity. 2012 May 25; 36(5): 705–716. doi:10.1016/j.immuni.2012.05.008; Immunity. 2019 June 18; 50(6): 1498–1512.e5. doi:10.1016/j.immuni.2019.04.010) is expressed mainly on lymphocytes and not on macrophages. This aspect should be considered and corrected

To which CXCL –type the authors do refer on line 438? Since there SMM is discussed one think that CXCR3 ligands are involved, but these cannot stimulate macrophage rtecruitment. Please make clear this paragraph and/or quote adequate bibliography.

Page 10, line 438

There is no evidence here that the CXCL-upregulation is dependent on malignant cells. This event could be induced by other infiltrating cells.

Author Response

Reviewer #1

Comments to the Author

General comments:

This paper evaluates the difference in gene expression between two histologic subtypes of malignant mesothelioma, that is sarcomatoid and epithelioid. They find significant differences as far as genes involved in either innate or acquired immune defense mechanisms are regarded. The findings they report are undoubtedly of value and deserves to be published. However, apart from the various comments reported below, two main observations should be considered.

Authors Comments: We thank the reviewer for the time and effort to carefully review our manuscript. We are glad about the positive feedback and we did our best to address all mentioned points, thereby hoping to significantly improve our presented results.

  1. The normal mesothelium is known to play a key role in defense mechanism, through ingestion of different particles and microorganisms, antigen presentation, ROS production and cytokine secretion (The International Journal of Biochemistry & Cell Biology 36 (2004) 9–16; Respiration 2008;75:121–133 DOI: 10.1159/000113629. Therefore it is not a surprise that neoplastic cells in SMM could maintain some features of normal cells while EMM do not. Perhaps their absence in EMM may be important. It should be of great interest to analyze the expression of genes involved in innate and acquired defense in normal cells for comparing them with those of SMM and EMM. This aspect could be at least discussed.

Authors Comments: We thank the reviewer for thoughtful suggestion. Of course, there are some limitations of our study, including the lack of verification of immune cell infiltration pattern via IHC or similar, the relatively small samples sizes, as a consequence of the low number of SMM available with more or less equal group distribution, and also the above mentioned point of the role of the mesothelium as an highly immunogenic surface with the lack of normal mesothelium for comparison (since this is extremely hard to obtain). We tried to address all those in a separate paragraph at the end of the discussion section.

  1. Conclusions must be limited to underline the differences in gene expression. Suggesting that in SMM, TAM and APC recruitment, phagocytosis and antigen processing and presentation could modulate local immune response is excessively speculative (considering the poor prognosis of this histotype).

The same comment is true for EMM. Therefore a more adequate concluding remarks are required.

Authors Comments: We agree with the reviewer and want to apologize for the inaccuracy and this highly speculative conclusions. We have revised our conclusions section and focused more on the differences directly measured rather than on those speculations drawn.

Specific comments:

Page 1, line 33

I think that the acronim used for Formalin Fixed Paraffin Embedded (FFPE) should be explained.

If you indicate If you report: twelve of those samples……..please continue to indicate ..ten epithelioid

Authors Comments: We totally agree with the points raised by the reviewer and revised the above mentioned points as indicated.

Page 2, line 57

…setting

Authors Comments: We thank the reviewer for careful revision and corrected the referenced typo.

Page 2, lines 86-111

In these paragraphs the authors briefly describe the cellular components known to form the MPM micro-environment. However at this point a comparison between the cell types present in SMM and those present in EEM should be important and may also result useful to interpretate the findings. The description as reported suggest that the same cell infiltrate seems to be present in both the MPM histotypes. Only a high level of MDSC   linked to a poor prognosis in EMM is cited.

Authors Comments: We are d’accord with the reviewer, that this is not clearly specified in the mentioned paragraph. Unfortunately, most studies did not distinguish between EMM and SMM in this context, and subgroup analysis are based on quite small sample sizes making conclusions not evident. We tried to clarify this at the end of the section.

Page 6, Fig. 2

Fig. 2 is very difficult to analyze. I suggest, if possible to include B-F as single pictures in supplemental materials and leave only A in fig. 2

Authors Comments: We are in line with the reviewer and split the figure as indicated.

Page 7, line 243

It is not correct to include TfR among the genes involved in the phagocytic process. The complex Tf-TfR is endocytosed through a chlatrin mediated mechanisms and, following acidification inside the cell, the vesicle recycles to the membrane making free again apotransferrin. Therefore I suggest to omit TfR from the list of genes involved in the ingestion process.

Authors Comments: We thank the reviewer for this observation and have omitted TfR from this paragraph.

Page 7, line 245

…… NADPH…..

Authors Comments: We want to apologize for the typo and corrected it.

Page 10, lines 406-449

 Resolving the complexity of chemokines network in a simple scheme is very difficult. For example following the chemokine/chemokine receptor guide of 2018 (The FEBS Journal 285 (2018) 2944–2971 ª 2018 ), it appears that CXCR1/2 is be involved in neutrophil recruitment. Are neutrophils recruited in EMM? Neutrophils are very citotoxic cells and this feature could be in principle beneficial for the host.

Conversely CXCR3, accordingly to recent published papers, (Immunity. 2012 May 25; 36(5): 705–716. doi:10.1016/j.immuni.2012.05.008; Immunity. 2019 June 18; 50(6): 1498–1512.e5. doi:10.1016/j.immuni.2019.04.010) is expressed mainly on lymphocytes and not on macrophages. This aspect should be considered and corrected

Authors Comments: We thank the reviewer for raising this very interesting point. Unfortunately, we have no conclusive evidence for the presence of neutrophils in our EMM. As we totally agree with the reviewer that the reduction of the chemokine network, especially but not limited to the CXC family, to that simplification is difficult and may lead to false assumptions, we forewent to draw this speculative conclusion of neutrophils play this major role here or which cell type in particular express CXCR1,2 or 3 and focussed on a more descriptive analysis of signals received from different malignancies. Nevertheless, we think that the mentioned mechanism of neutrophil recruitment as anti-cancer cytotoxic cells is very interesting, especially in the context of upcoming ICB in MPM, we think this should be addressed in a further, separate study in detail.

To which CXCL –type the authors do refer on line 438? Since there SMM is discussed one think that CXCR3 ligands are involved, but these cannot stimulate macrophage rtecruitment. Please make clear this paragraph and/or quote adequate bibliography.

Page 10, line 438

There is no evidence here that the CXCL-upregulation is dependent on malignant cells. This event could be induced by other infiltrating cells.

Authors Comments: We totally agree with the reviewer and want to apologize for the inaccuracy and this highly speculative conclusions drawn. As mentioned in our last comment, reduction of the chemokine network to that simplification is difficult and may lead to false assumptions, and want therefore to refrain from this speculative conclusion. We deleted the last part of this paragraph and hope to get more evidence in subsequent studies, including both in vitro and in vivo data.

Reviewer 2 Report

The manuscript on gene expression analysis in MPM describes a highly complex research work that analysed RNA from 22 samples of sarcomatoid and epithelioid MPM, equally represented. 698 immune-related genes under appropriate quality control were evaluated. Distinctive expression patterns in immune responses were recognised in EMM and SMM tumor microenvironment. Authors made a great effort to explain prominent signal pathways and hypothesised a possible benefit of PDL-1 therapy. Although the number of included samples was relatively small, results support the conclusion, that immune responses in tumor microenvironment of EMM and SMM are distinctive and probable require specific therapeutic approaches.

Authors mentioned radiological and clinical data of patients that donated histology samples to the study. I wonder if any correlations between gene expression pattern, response to therapy and survival were found?

Author Response

Please see att.

Round 2

Reviewer 1 Report

The authors have addressed all the reviewer's comments in an acceptable way

Best regards